# Three-way Calibration Checks Using Ground-Based, Ship-Based and Spaceborne Radars

Alain Protat [1], Valentin Louf [1], Joshua Soderholm [1], Jordan Brook [2], William Ponsonby [3]

[1] Australian Bureau of Meteorology, Melbourne, Australia

[2] University of Queensland, Brisbane, Australia

[3] Engineering and Technology Program, CSIRO National Collections and Marine Infrastructure, Hobart, Australia

*Correspondence to*: Alain Protat (alain.protat@bom.gov.au)

**Abstract.**

This study uses ship-based weather radar observations collected from *Research Vessel Investigator* to evaluate the Australian weather radar network calibration monitoring technique that uses spaceborne radar observations from the NASA Global Precipitation Mission (GPM). Quantitative operational applications such as rainfall and hail nowcasting require a calibration accuracy of ±1 dB for radars of the Australian network covering capital cities. Seven ground-based radars along the western coast of Australia and the ship-based OceanPOL radar are first calibrated independently using GPM radar overpasses over a 3-month period. The calibration difference between the OceanPOL radar (used as a moving reference for the second step of the study) and each of the 7 operational radars is then estimated using collocated, gridded, radar observations to quantify the accuracy of the GPM technique. For all seven radars the calibration difference with the ship radar lies within ± 0.5 dB, therefore fulfilling the 1 dB requirement. This result validates the concept of using the GPM spaceborne radar observations to calibrate national weather radar networks (provided that the spaceborne radar maintains a high calibration accuracy). The analysis of the day-to-day and hourly variability of calibration differences between the OceanPOL and Darwin (Berrimah) radars also demonstrates that quantitative comparisons of gridded radar observations can accurately track daily and hourly calibration differences between pairs of operational radars with overlapping coverage (daily and hourly standard deviations of ~ 0.3 dB and ~ 1 dB, respectively).

## 1 Introduction

Operational radar networks play a major role in providing situational awareness and nowcasting in severe weather situations, including heavy rain, flash floods, hailstorms, and wind gusts. Such radar-based information is then used by forecasters as guidance for issuing severe weather warnings. The quality of these radar-derived products in real-time is driven to a large extent by how well the underlying radar measurements are calibrated. Recently, the Australian Bureau of Meteorology (BoM) has developed an operational radar calibration framework to monitor the calibration of all BoM operational radars in real-time (Louf et al. 2019, hereafter L19). This approach is based on a combination of three techniques. The objective of this technique is to achieve an absolute calibration accuracy better than 1 dB, which is the operational calibration requirement in Australia for quantitative use of the Australian weather radar observations over capital cities (so-called Tier 1 radars). At the heart of this framework lies the so-called Volume Matching Method (VMM), initially developed by Schwaller and Morris (2011) and further

improved by Warren et al. (2018, hereafter W18). In this VMM technique, intersections between individual ground-
based radar beams and NASA Tropical Rainfall Measurement Mission (TRMM, Simpson et al. 1996) or Global
Precipitation Mission (GPM, Hou et al. 2014) scanning Ku-band radar beams are averaged over an optimally
defined common sampling volume (see W18 for more detail). In what follows, we will use the term "calibration" to
refer to calibration differences between ground or ship-based radars and the GPM radar taken as the "reference".
However, it must be noted that reflectivities measured by the GPM radar are not a normed reference, which implies
that our use of the term "calibration" is strictly not correct.

A major advantage of using the GPM VMM technique is that the spaceborne radar provides a single source
of reference to calibrate all radars of an operational network. This was also well demonstrated in Kollias et al.
(2019) in the context of calibrating the U.S. Atmospheric Radiation Measurement (ARM) cloud radar network using
the spaceborne CloudSat radar. Despite multiple possible sources of errors contributing to the VMM calibration
error estimate, such as temporal mismatch, imperfect attenuation corrections, gridding and range effects, and
differences in radar minimum detectable signal, the overall accuracy of such technique is thought to be better than 2
dB for individual overpasses (Schwaller and Morris, 2011; W18; L19. It must be noted however that there has been
no independent quantification of this accuracy. This is the main objective of this study, where we use dual-
polarization C-band weather radar (OceanPOL) observations collected on board the Marine National Facility (MNF)
Research Vessel (RV) Investigator between Darwin and Perth, Australia, as part of the *Years of the Maritime*
*Continent – Australia* (YMCA, Protat et al. 2020) and the *Optimizing Radar Calibration and Attenuation*
*corrections* (ORCA) experiments to evaluate the approach of calibrating a whole radar network using GPM. The
concept of this study is presented in Fig. 1. GPM observations are first used to calibrate both the ship-based radar
and all the operational ground-based radars along the western coast of Australia independently. The ship-based radar
observations calibrated using GPM are then individually compared with those from each ground-based radar as the
ship sails close to them. Since all radars (including OceanPOL) have been calibrated using GPM, the differences
between ship-based and ground-based observations can be interpreted as an error estimate of the GPM calibration
technique, with some unknown additional contribution from errors due to the ship-ground radar comparisons
themselves. These errors coming from ship-ground comparisons are expected to be much lower than those arising
from the GPM / ground radar comparisons. Indeed, the advantage of using a ship-based radar relative to a
spaceborne radar is that many of the error sources in ground-based / satellite radar comparisons are reduced to a
minimum. Taking advantage of a month-long dataset of calibration difference estimates between OceanPOL and the
Darwin radar, we also assess the operational potential of daily and calibration change monitoring using overlapping
ground-based radar observations.

The remainder of this paper is organized as follows. In section 2, we briefly describe the YMCA and
ORCA experiments, the characteristics of radars used in this study, and the calibration techniques. In section 3, we
present the main findings of this study. Concluding remarks are presented in section 4.
**2 Radar observations during YMCA and ORCA and calibration comparisons**

In this section, we briefly introduce the datasets collected during the YMCA and ORCA experiments, the
details of all radars involved in this study, and the techniques used to calibrate the ground and ship radars with the
spaceborne radar and to compare ground and ship radars.

## 2.1 The YMCA and ORCA experiments

*RV Investigator* OceanPOL radar observations used in this study were collected as part of two back-to-back field experiments. The first experiment is the Australian contribution to the Years of the Maritime Continent (YMCA), which is an international coordinated effort to better understand the organization of coastally induced convection over the Maritime Continent and its complex interactions with large-scale drivers, with the ambition to better represent these processes in global circulation models characterized by large and persistent rainfall biases. During the second phase of YMCA (12 November – 19 December 2019), the sampling strategy was to position *RV Investigator* off the coast around Darwin in a dual-Doppler configuration with either the Warruwi (north-east of Darwin) or Berrimah (Darwin) operational C-band Doppler radars to characterize the rainfall, morphological, and dynamical properties of convective systems developing near the coast and propagating offshore, which are particularly poorly forecasted in this region (e.g., Neale and Slingo, 2002; Nguyen et al. 2017a,b), but are thought to contribute about half of the rainfall along tropical coasts (e.g., Bergemann et al. 2015). In this study, we also take advantage of the month-long time series of OceanPOL – Berrimah radar observations to quantify the variability of radar calibration on daily and hourly timescales.

The second field experiment (ORCA) was conducted during a transit voyage to relocate *RV Investigator* from Darwin to Perth, Western Australia. This transit voyage was an ideal opportunity to collect collocated radar samples with several operational radars along the coast (Fig. 1). Specific stops of three hours were scheduled in the vicinity of each radar in the event of precipitation within range of OceanPOL and of the ground-based radar. Of the eight possible radars, we have luckily been able to collect such collocated precipitation samples for six of them, except Geraldton and Carnarvon. In this study we will use all these collocated samples to quantify how well the calibration estimate provided for each radar by the GPM technique agree with the calibration estimates obtained using OceanPOL as a second and more accurate source of reference.

## 2.2 The radars of this study

Table 1 summarizes the relevant information about all radars used in this study. The Australian radar network comprises a large variety of radars from different generations, frequencies (although radars in this study are all C-band radars, other parts of the country are covered by S-band radars), beamwidths (ranging from 1.0° to 1.7°), range resolutions (ranging from 250m to 1000m), and total time to complete each volumetric sampling (from 6 min for more recent radars to 10 minutes for older radars). Several radars of the network are installed in very remote locations, bringing specific challenges for the regular maintenance and return to service in case of hardware failure. As a result, maintaining an accurate calibration of this network is more difficult than in other countries. At the time of the YMCA and ORCA experiments, all radars operated continuously. The Berrimah (Darwin) and Serpentine (Perth) radars are Tier 1 radars (as they cover capital cities), while all other radars in Table 1 are Tier 2 radars. Tier 1 and 2 radars have a calibration accuracy requirement of better than 1 and 2 dB, respectively. The internal calibration accuracy of these operational radars is ideally checked six-monthly by BoM radar engineers as part of their routine maintenance. However, periods between visits can be longer for radars in remote locations. The calibration check only includes measurements of gains and losses at different check points of the transmission and reception chains. No end-to-end calibration using external targets is ever performed. Special visits to sites are organized when a radar

is down or when complaints are issued by the public about radar data quality. The extensive recommendations
outlined in Chandrasekar et al. (2015) have not been implemented for the Australian radar network yet.
The GPM KuPR and OceanPOL radars are the most modern radars. It must be noted that the OceanPOL
radar is the only dual-polarization radar. This important feature for several applications is not used in the present
study, except for the quality control of the OceanPOL radar data. A critical aspect of operating a radar on a research
vessel is the need to compensate for ship motions and velocity in real-time. To do so, the OceanPOL antenna control
system ingests the real-time inertial motion unit data from the ship at 10 Hz and steers the radar beam in real-time in
the requested azimuth and elevation direction. The accuracy of this stabilization has been found to produce a
pointing accuracy better than $0.1°$, even in harsh sea conditions. Doppler measurements are automatically corrected
in real-time for the Doppler component induced by ship velocity components. Dual-polarization moments are also
corrected using the statistical corrections proposed in Thurai et al. (2014). The same calibration procedure as that
employed by BoM is used for OceanPOL (internal measurements of gains and losses, no end-to-end calibration),
which does not include the calibration recommendations from Chandrasekar et al. (2015).
As discussed previously, the GPM Ku-band radar measurements are considered as the reference for the
calibration of all radars in this study. The GPM radar calibration procedure, described in detail in Masaki et al.
(2020) inherited from years of calibration work undertaken as part of the previous satellite radar mission, the
Tropical Rainfall Measurement Mission (TRMM). This calibration comprises an internal calibration (monitoring
closely the gains and losses of each component of the radar) and an external calibration procedure using a ground-
based calibrator and sea surface of well-known backscatter. Importantly, the GPM mission also benefits from
extensive field experiments undertaken as part of the Ground Validation program, including in-situ ground and
aircraft validation of the products of the GPM mission. By comparing different approaches for the GPM Ku-band
radar calibration, Masaki et al. (2020) demonstrated that the accuracy of the radar was well within the $\pm 1$ dB
requirement. In our study, Version 5 of the GPM 2AKu product has been used for all comparisons in this study
(Kidd et al. 2017), which includes the latest calibration from Masaki et al. (2020) and contains attenuation-corrected
Ku-band reflectivities. GPM attenuation correction is achieved using a hybrid approach combining the traditional
Hitschfeld - Bordan technique (Hitschfeld and Bordan, 1954) and the so-called Surface Reference Technique
(Meneghini et al., 2004). To compare GPM Ku-band radar with C-band radars in this study, all GPM Ku-band
reflectivities have been converted to their equivalent C-band reflectivities using Eq. 5 in L19.
**2.3 The S³CAR radar calibration framework**
Recently, BoM has developed the operational S³CAR (Satellite, Sun, Self-consistent, Clutter calibration
Approach for Radars) framework to monitor the calibration of the BoM operational radars in real-time (operational
version of L19). This approach is based on a combination of three techniques. The first technique, the Relative
Calibration Adjustment (RCA, e.g., L19; Wolff et al. 2015), assumes that the $95^{th}$ percentile of "ground clutter"
radar reflectivities (buildings, topographic structures, trees, etc …) within 10 km range is constant. This technique
tracks changes in daily calibration to better than 0.2 dB (L19) but does not provide an estimate of the absolute
calibration. The second technique (W18) statistically compares collocated ground radar and spaceborne Ku-band
radar from the NASA TRMM (1997-2014) and GPM (2014-present) missions. The operational implementation of
the GPM calibration technique closely follows the description given in W18. Satellite and ground-based radar
observations are first matched to a common volume. We require at least a minimum of 10 satellite profiles within
the ground radar domain to select and process a satellite overpass. The melting layer is detected by the operational
GPM algorithms and excluded from the matched volumes due to uncertainties in frequency conversions for melting
hydrometeors. Matched volumes in both liquid and ice phases are retained (like in W18). Non-uniform beam filling
effects of the matched volumes are mitigated by only selecting volumes that are 95% filled. A maximum ground-
based reflectivity threshold of 36 dBZ is used in the analysis of matched volumes to mitigate the potential impact of
attenuation correction errors.
From our experience, and as reported in L19, this technique provides an absolute calibration with an
accuracy of about 2 dB from each overpass. The S$^3$CAR framework uses the RCA technique to detect stable periods
of calibration and averages calibration estimates from all GPM overpasses within each period, improving the
absolute calibration accuracy, hopefully to better than 1 dB. Note that these values of 2 dB and 1 dB are qualitative
error estimates based on visual inspection of the variability of calibration error estimates from successive satellite
overpasses. The third technique used in S$^3$CAR is the solar calibration technique, which is a faithful implementation
of the Altube et al. (2015) method, with additional corrections for a possible levelling error of the radars as
described in Curtis et al. (2021). The solar calibration technique uses sun power measurements collected at the
Learmonth observatory, Western Australia. This technique is mostly used in conjunction with the RCA and GPM
outputs to diagnose whether a change in calibration is due to the transmitting chain (RCA and GPM detect a change
but not the solar calibration technique) or the receiving chain (all techniques detect a change). This is an important
diagnostic to help radar engineers troubleshoot a radar issue and enable rapid return to service.
The BoM does not operate a disdrometer network. As a result, the technique outlined in Frech et al. (2017),
which compares disdrometer simulations of reflectivity with measured radar reflectivities cannot be added to the
S3CAR framework. In the future, with the increasing number of dual-polarization radars in the Australian network,
we are planning to investigate the benefits of the so-called self-consistency of polarimetric variables and may add
this technique to the framework.
Among all operational radars considered in this study, only two of these radars (Berrimah and Geraldton)
send the unprocessed reflectivities to Head Office in real-time, allowing for the full S$^3$CAR process to be used to
calibrate these radars. The term "unprocessed" here refers to radar data still containing noise and all typical radar
signal contaminations, including ground clutter and sun spikes used in our calibration techniques. For the other
radars, post-processing is done on-site to reduce the bandwidth required to send the radar data in real-time (these
radars are in very remote places). As a result, ground clutter and sun interference have largely been removed for
these radars, which implies that only the GPM part of the S$^3$CAR framework can be used. As explained, this reduces
the accuracy of the calibration estimate for such radars.
**2.4 Statistical comparisons between OceanPOL and the ground radars**
Calibration between ground-based radars and OceanPOL proceeds by first gridding observations from each
radar to a common 1 km horizontal / 500 m vertical resolution domain, then building a joint frequency histogram of
reflectivity values from all common grid points. The expectation from such plots is that they should exhibit a
systematic shift, corresponding to a difference in calibration between the two radars, with a large amount of
variability in these comparisons owing to all the sources of errors involved in such comparisons (differences in exact
time of observations of a grid, imperfect attenuation corrections, gridding artefacts, differences in implicit resolution
of radar volumes at different ranges, differences in minimum detectable signal …). The gridding technique used for
all radars is the same and follows Dahl et al. (2019). This gridding technique uses a constant radius of influence
(3.5km) and a weighted summation with distance to the centre of the grid for points belonging to the same elevation
angle but a linear interpolation in the vertical using data from the elevations below and above each grid. This
technique has the advantage of not producing the typical artificial vertical spreading of observations below / above
the lowest / highest elevation angles observed when using a radius of influence in all directions. Depending on how
old the ground radars are, different minimum reflectivity thresholds are used in the comparisons to mitigate potential
artefacts in calibration difference estimates due to the degraded sensitivity and reflectivity resolution of the older
radars for low to intermediate reflectivities. In general, a relatively high threshold of 20-25 dBZ was required, which
also had the advantage of reducing the potential impact of different non-uniform grid filling at the edges of the
convective systems due to different radar detection capabilities.

OceanPOL data have been corrected for attenuation using the Gu et al. (2011) C-band dual-polarization

technique available in the Py-ART toolkit (Helmus and Collis, 2016). The operational radars have been corrected for
attenuation using C-band reflectivity – attenuation relationships derived from the OceanRAIN dataset (Protat et al.
2019). It must be noted that additional comparisons done without attenuation corrections of the ground radars did
not yield large differences (less than 0.5 dB in all sensitivity tests conducted). This is presumably due to the fact that
there are many more points below 30-35 dBZ than above in those comparisons, resulting in a relatively minor
impact of attenuation on these statistical comparisons. Also, the ship and ground radars were generally not far away
from each other (typically 20-40 km), so the viewing geometry of the storms was quite similar from both radars in
most cases, resulting in similar levels of attenuation along the two different paths through the storms.

The scanning sequence employed for OceanPOL uses the exact same 14 elevation angles used throughout

the operational radar network. The start of each OceanPOL scanning sequence is synchronized with that of the
operational radars running a 6-minute sequence (starts on the hour then every 6 minutes), which implies that
temporal differences in volumes sampled by OceanPOL and the radars running the 6-minutes sequence are minimal.
The impact of temporal evolution on the comparisons between OceanPOL and the radars running a 10-minute
sequence will naturally be larger. To minimize this impact in our comparisons, we have discarded files for which the
start time differs from the OceanPOL start time by more than 2 min.

Finally, to mitigate the potential impact of wet radome attenuation at C-band on the comparisons, we have

screened out observations where precipitation was present within 5km of either of the radars from the comparisons.
More precisely, for each volumetric scan we estimate the precipitation fraction within 5 km, and if more than 20%
of this area is covered with precipitation, we conservatively discard this scan. However, it must be noted that results
obtained when changing that threshold were very similar, with maximum statistical differences in estimated
calibration difference less than 0.3 dB (not shown). From a visual inspection of radar scans, we inferred that this was
due to rainfall generally not observed over and around the radars when such comparisons were made.



## 3 Results

In this section, we present the main results of this three-way calibration comparison exercise. Comparisons between OceanPOL and the ground-based radars, all calibrated using GPM, are used to quantify the accuracy of the GPM VMM technique. The day-to-day variability of ground – ship radar comparisons over a month is also used to quantify the accuracy of daily calibration monitoring using overlapping ground-based radars and its potential for operational use. Lastly, we explore the potential for tracking calibration differences at the hourly time scale rather than the daily time scale using overlapping ground-based radars.

### 3.1 The accuracy of the GPM VMM technique

As illustrated in Fig. 1, the first part of the calibration consistency check is to calibrate OceanPOL and the ground radars using the same single independent source, the GPM spaceborne radar. All calibration results are summarized in Fig. 2. We are fortunate enough that over two months including the YMCA and ORCA observational periods, the rainfall activity allowed us to collect a reasonable number of GPM overpasses over each radar (except for Learmonth, radar 29, Fig. 2). As a result, for radar 29, we will use an older calibration estimate (-2.6 dB), derived from a GPM overpass with many matched volumes in July 2019 and will assume that its calibration has not changed. As discussed previously, the RCA technique can be used to accurately track changes in calibration. Unfortunately, among all radars included in Fig. 2, the RCA can only be applied to radar 63. Additional checks of the outputs of the RCA technique for radar 63 (not shown) indicated that the calibration of radar 63 had not changed over that period, which means that we can simply average all the estimates of calibration error from individual overpasses to come up with a more accurate estimate for this radar 63. Although the RCA technique cannot be used for the other radars, some insights into the calibration stability can be gained from individual calibration estimates from individual GPM overpasses in each panel of Fig. 2. Considering the expected typical error of 2 dB for individual GPM overpasses as a guideline, it seems reasonable to assume that the calibration of the OceanPOL, Warruwi (77), Dampier (15), Broome (17), and Serpentine (70) radars has not changed over the observational period either, with fluctuations around the mean calibration error estimate less than ~1.5 dB. Results using the solar calibration technique for OceanPOL also indicate that the OceanPOL receiver calibration has remained constant, to within 1 dB, over the study period (sun power of about -93 dBm). The Port Hedland (16) radar is more problematic, as the time series shows calibration error estimates ranging from -8 dB to -2.5 dB over that period. However, the three overpass points closest to the date when collocated observations with OceanPOL were collected (26 December 2019) seem to agree reasonably well (around the mean value of -5 dB), so we will use this value of -5 dB in the following but will keep in mind the lower confidence in this calibration figure.

The final step of this calibration consistency check study consists in using the OceanPOL radar (previously calibrated using GPM, Fig. 2) as a second moving reference to compare with the ground-based radars. As explained earlier, satellite – ground comparisons are characterized by multiple sources of errors, including differences in sampled volumes (although great care is taken to match sampling volumes as accurately as possible, e.g., Schwaller and Morris 2011, W18, L19), non-uniform beam filling effects, temporal mismatch between observations, differences in minimum detectable signal, and radar frequency differences requiring conversion (most problematic in the melting layer and ice phase of convective storms where this correction is more uncertain, see W18). In comparison, ship radar – ground radar comparisons, especially when radars are, as in this study, reasonably close to

each other to minimize differences in sampling volumes, are less prone to all these errors. The radar frequency is the
same. The sampling volume and temporal mismatches are also expected to be less problematic (but not entirely
negligible, especially for the radars running a 10-min sequence, see discussion in section 2.4). These more accurate
ship – ground radar comparisons should therefore be considered as an indirect evaluation of the GPM validation
technique and if successful, a demonstration of the value of using such GPM data as a single source of reference for
the calibration of a whole national network as is done in Australia with $S^3$CAR.

Figure 3 shows an example of the 2D frequency histograms of reflectivity that are used to estimate
calibration differences between OceanPOL and any of the radars. This particular figure is for the Berrimah radar
(63) for one day (21 November 2019) of the YMCA experiment. Such frequency distribution plots can be
normalized in two different ways. If the number of points in each reflectivity pixel is divided by the total number of
points (as in Fig. 3a), it highlights where most of the comparison points are in the reflectivity – reflectivity space,
and therefore what contributes most to the mean calibration difference estimate. When the number of points in each
pixel is divided by the total number of points in each reflectivity bin on the x-axis (Fig. 3b), the joint distribution
provides a better visual sanity check of the systematic shift of the joint distribution produced by the calibration
difference over the whole reflectivity range and allows detection of other potential artefacts. In the example of Fig.
3a, which is typical of all comparisons made in this study, it is clear that reflectivities less than 35 dBZ contributed
most to the estimation of the mean calibration difference of 0.9 dB between the two radars. On another hand, Fig.
3b shows more clearly that there is indeed a consistent shift in reflectivity values across the whole reflectivity range,
as expected from a (systematic) calibration difference. An important feature of Fig. 3 is the observed large
variability around the mean calibration difference. The standard deviation of calibration difference for all
comparisons in this study was typically between 4 and 6 dB. It must be noted that this large standard deviation is an
estimation of the errors on calibration difference of each individual pixel, not that of the daily estimate. The higher
number of days spent collecting collocated observations off the Berrimah (63) and Warruwi (77) radars also offers
an opportunity to estimate daily calibration differences and take a closer look at the day-to-day variability of
calibration differences.

When including all days of observations for radars 63 and 77 (25 days for radar 63 and 4 days for radar 77
with precipitation), the mean calibration difference between OceanPOL and radars 63 and 77 are 0.4 dB and -0.3
dB, respectively (see Fig. 4 for radar 63, Fig. 5a for radar 77, see also Table 2 for a summary of all calibration
differences found in this study). The other relatively recent, better-quality operational radar included in this study is
radar 70 (Perth). For this radar, only short duration drizzle and scattered showers were observed when *RV*
*Investigator* approached its destination (Fremantle port), resulting in less points for the calibration difference
estimate. Despite the short duration dataset for radar 70, the 2D joint histogram of reflectivities show a consistent
difference across the whole reflectivity range, with a mean calibration difference of -0.4 dB (Fig. 5f). These three
estimates are well below the required accuracy of 1 dB for operational applications, which indicates that for these
four good-quality radars (OceanPOL and radars 63, 77, and 70), the GPM comparisons provided a consistent
calibration to within ± 0.5 dB. However, those are the comparisons where errors were expected to be smallest, given
the large number of days included in the comparisons for radars 63, and the excellent synchronization of the 6-min
scanning sequences with OceanPOL for these three radars.

Let us now turn our attention to the quantitative comparisons between OceanPOL and the older operational
radars (15, 16, 17, 29) running with a 10-minute scanning sequence and/or a degraded range resolution (as reported
in Table 1), and only a few opportunistic hours of collocated samples with precipitation (see list of time spans in
Table 2). Visual inspection of gridded radar data revealed the presence of strong anomalous propagation (AP) signal
in the lower levels (up to about 2km height ASL) for radars 15, 16, and 29, which has not been filtered correctly by
the operational radar post-processing suite. This problem is well known to the BoM forecasters. As a result, for these
radars, two sets of results are presented in Table 2. Calibration differences obtained from all data are labelled "AP"
and those obtained when screening out all common grids below 2km height are labelled "noAP". Figure 5 shows the
2D joint histograms of reflectivity when the anomalous propagation is screened out. The largest impact of
anomalous propagation is found for radar 16, with a difference of 0.9 dB between estimates with and without AP
screening. For the two other radars 15 and 29, the impact is modest (0.3 to 0.5 dB). This is due to the higher
proportion of samples located below 2 km height for the radar 16 case (not shown) than for the two other cases.
Overall, this result is shown to illustrate that particular attention needs to be paid in regions prone to anomalous
propagation effects. From Table 2 and Fig.5, the calibration differences with OceanPOL for these older radars are
+0.3 dB (radar 15), +0.1 dB (radar 16), +0.4 dB (Broome, radar 17), and +0.1 dB (radar 29). In summary, all seven
radars considered in these comparisons are characterized by calibration differences with OceanPOL within +-0.5 dB,
despite the large variability in radar quality and number of samples included in the calibration difference estimates
(reported in Fig. 5). As a result, we can safely conclude that these comparisons validate the concept of using the
GPM VMM calibration technique as a single source of reference to accurately calibrate and monitor calibration of
national radar networks.
**3.2 The accuracy of daily calibration monitoring from overlapping ground-based radars**

As introduced earlier, the day-to-day variability of calibration differences between ship and ground-based
radars can be analysed using the month of collocated samples between OceanPOL and the Berrimah radar collected
during YMCA (coloured points in Fig. 4). From Fig. 4, some simple statistics can be derived and discussed. The
minimum and maximum calibration differences over the month-long time series are -0.2 and +1.1 dB, which
corresponds to minimum and maximum differences of -0.6 and +0.7 dB around the mean value of 0.4 dB. The
colour of the points is the number of samples that were available to estimate the daily calibration difference. The
coloured error bars are estimates of the hourly standard deviation of calibration difference for each day. From a
close inspection of the location of points with respect to the mean value for the period, there does not seem to be any
obvious relationship between the number of points and how close the estimates are to the mean value of 0.4 dB. This
result shows that the number of samples is not the main source of differences between daily estimates.

The standard deviation of daily calibration difference between Berrimah and OceanPOL over this month of
data is 0.33 dB (Fig. 4). Since this standard deviation value includes any potential natural variability of the daily
calibration difference and the variability due to uncertainties in these daily ship – ground radar comparisons such as
spatial resolution differences and temporal mismatches, this value of 0.33 dB can be considered as an upper bound
for the uncertainty in daily calibration difference estimates. To check whether the natural variability of daily radar
calibration was minimal over that month of Darwin observations, we have added in Fig. 4 the time series of daily
mean RCA values (black points) used as part of our operational S[3]CAR calibration monitoring technique as another
calibration variability metrics. It has been shown that this RCA technique could track changes in daily calibration to
better than about 0.2 dB (L19). To better compare variabilities obtained from calibration differences and the RCA,
we have subtracted the mean RCA (54.11 dBZ) value to each daily RCA value and added the mean calibration
difference over the whole period (0.4 dB), so that the daily RCA time series is centred on the mean calibration
difference (blue line). Over this whole period, the standard deviation of the RCA value is 0.12 dB, which confirms
the L19 results. This standard deviation is smaller than that of the OceanPOL – Berrimah comparisons (0.33 dB). If
we assume that the standard deviation of the RCA value is an upper bound for the natural variability of the daily
calibration figure, this result shows that most of the variability in calibration difference between the OceanPOL and
Berrimah radars (0.33 dB) is in fact a measure of the inherent uncertainties of gridded radar comparisons. This
important result highlights that such quantitative comparisons of overlapping gridded radar observations can be
successfully used to monitor the consistency of daily calibration of operational radars with overlapping coverage to
better than the 1 dB requirement.

### 3.3 The accuracy of hourly calibration monitoring from overlapping ground-based radars

The last thing we explore with this Darwin dataset is the potential for tracking calibration differences at the
hourly time scale rather than the daily time scale. To do so, for each day of observations, we have estimated the
calibration difference from 1-hour chunks of collocated data, then estimated the standard deviation of the hourly
estimates for each day. An example of such daily analysis is shown in Fig. 6 for a day (08/12/2019) where 15
successive hours of collocated samples were available. Although this example includes more hours of comparisons
than most other days, it is very typical in terms of the hour-to-hour variability we observe each day, making it a
good candidate for illustrative purposes. We have not elected to screen out hours with fewer points, which, as can be
seen from hours 14 and 15, would have resulted in a lower hourly standard deviation for that case. This should
probably be done in an operational implementation. In this respect, the standard deviation of hourly calibration
difference presented in Fig. 4 can be considered as an upper bound for the hourly standard deviation. The hourly
standard deviation is shown in Fig.6 as a red error bar on top of the daily average point, and as a coloured error bar
over each daily average in Fig. 4. Over the 1-month study period, the average hourly standard deviation derived
from all hourly estimates is 0.8 dB, which is within the 1 dB requirement, but the two extreme values are 0.5 and 1.5
dB (Fig. 4), indicating that occasionally the hourly estimates of calibration difference would not fully meet this
requirement. From Fig. 4, it also appears that there is no inverse relationship between the number of samples and the
hourly standard deviation, which could have perhaps been expected. For instance, the two points with highest hourly
standard deviation (02 and 06 December 2019) are at both ends of the number of samples spectrum, and the three
points with the lowest hourly standard deviations are in the lower half of the number of samples spectrum. Fig.4 also
shows that when using the hourly standard deviation as an error bar, the mean value over that period (0.4 dB) is
always included within one standard deviation of the daily estimate. These results would obviously need to be
confirmed with more observations in the future but do highlight the potential for hourly tracking of calibration
differences, enabling very early detection of issues with operational radars.

## 4 Conclusions

In this study, we have used collocated observations between spaceborne, ship-based, and ground-based radars collected during the YMCA (off Darwin) and ORCA (transit voyage between Darwin and Perth) experiments to gain further insights into the suitability and accuracy of using spaceborne radar observations from the GPM satellite mission to calibrate national operational radar networks, and to assess the potential of using data from overlapping ground-based radars to track calibration changes operationally at the daily and hourly time scales.

A major advantage of the GPM VMM technique is that all radars of the network are calibrated against a single source of reference. The GPM VMM literature (Schwaller and Morris, 2011; W18; L19) suggests that errors are of about 2 dB from individual GPM overpasses to better than 1 dB when stable periods of calibration can be estimated using the RCA technique and individual GPM estimates can be averaged. However, these errors have never been fully quantified. Using collocated weather radar observations between the OceanPOL radar on *RV Investigator* and 7 operational radars off the northern and western coasts of Australia (all calibrated using GPM), we found that for all seven operational radars, the calibration difference with OceanPOL was within ±0.5 dB, well within the 1 dB requirement for quantitative radar applications (-0.3, +0.4, +0.4, +0.1, +0.3, +0.1, and -0.4 dB). This important result validates the concept of using the GPM spaceborne radar observations to calibrate national weather radar networks.

From the longer YMCA dataset collected when RV Investigator was stationed off the coast of Darwin for about a month, the day-to-day variability of calibration differences between the OceanPOL and Darwin (Berrimah) radars was estimated and compared with the daily calibration variability estimated using the RCA technique. From these comparisons, we found that the natural variability of daily radar calibration was small over our month of observations (~0.1 dB daily standard deviation). These comparisons also demonstrated that the intercomparison of gridded radar observations had the potential to estimate calibration differences between radars with overlapping coverage to within about 0.3 dB at daily time scale and about 1 dB at hourly time scale. Such technique will be added to our operational S³CAR calibration monitoring framework as an additional calibration monitoring reference between GPM overpasses when the RCA technique cannot be applied.

## Acknowledgments

The Authors wish to thank the CSIRO Marine National Facility (MNF) for its support in the form of *RV Investigator* sea time allocation on Research Voyages IN2019_V06 (YMCA) and IN2019_T03 (ORCA), support personnel, scientific equipment, and data management. Tom Kane and Mark Curtis from BoM are also warmly thanked for always patiently answering our relentless questions about the Australian weather radar network intricacies.

## Code availability

Codes developed for this study are protected intellectual property of the Bureau of Meteorology and are not publicly available.

## Data availability

All OceanPOL and Level 1b data from the operational radar network used in this study are available at
http://www.openradar.io. The NASA GPM radar data were obtained using the STORM online data access interface
to NASA's precipitation processing system archive (https://storm.pps.eosdis.nasa.gov).

**Sample availability**

**Sample availability**
No samples were used in this study.

**Author contribution**
AP, JS, VL, JB, and WP collected the datasets used in this study. VL produced the GPM comparisons using the
operational S3CAR technique. JS produced post-processed volumetric and gridded data for all ground-based radars.
VL produced the gridded OceanPOL data. JB developed the gridding technique used in this study. AP designed and
coordinated the YMCA and ORCA field experiments, analyzed the results, and wrote the manuscript. VL, JS, JB,
and WP provided edits of the manuscript.

**Competing interests:**
The authors declare that they have no conflict of interest.

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

**Tables**

| Radar ID or Platform | Name | Make | (lat, lon) | Band | $\omega$ (°) | $\Delta r$ (m) / $\Delta t$ (min) |
|---|---|---|---|---|---|---|
| GPM | KuPR | N/A | Variable | Ku | 0.7 | 125 / NA |
| RV Investigator | OceanPOL | DWSR-2501C-SDP | Variable | C | 1.3 | 125 / 6 |
| 15 | Dampier | WSR81C | (-20.654; 116.683) | C | 1.7 | 1000 / 10 |
| 16 | Port Hedland | TVDR2500-8 | (-20.372; 118.632) | C | 1.7 | 500 / 10 |
| 17 | Broome | DWSR2502C-8 | (-17.948; 122.235) | C | 1.7 | 500 / 10 |
| 29 | Learmonth | TVDR2500-8 (Digital upgrade) | (-22.103; 113.999) | C | 1.7 | 250 / 10 |
| 63 | Berrimah (Darwin) | DWSR2502C-14 | (-12.456; 130.927) | C | 1.0 | 250 / 6 |
| 70 | Serpentine (Perth) | TVDR2500-14 | (-32.392; 115.867) | C | 1.0 | 500 / 6 |
| 77 | Warruwi | DWSR2502C-14 | (-11.648; 133.380) | C | 1.0 | 250 / 6 |

Table 1: Main characteristics of the radars used in this study: radar ID in the operational radar network or platform,
name, make, coordinates, frequency band, beamwidth $\omega$ (°), range bin size $\Delta r$ (m), and total time to complete the
volumetric sampling $\Delta t$ (min). OceanPOL and all ground-based radars have been manufactured by the Enterprise
Electronics Corporation (EEC).

| Date | Time Span (UTC) | Radar | Calibration Error (Radar – OceanPOL) |
|---|---|---|---|
| 20191115 | 04:00 – 07:00 | 77 | -0.2 |
| 20191117 | 04:00 – 08:00 | 77 | +0.5 |
| 20191127 | 06:00 – 11:00 | 77 | -0.2 |
| 20191128 | 03:00 – 07:00 | 77 | -0.6 |
| All dates above | All time spans above | 77 | -0.3 |
| All dates in Fig. 4 | Miscellaneous | 63 | +0.4 |
| 20191225 | 12:00 – 21:00 | 17 | +0.4 |
| 20191226 | 18:00 – 24:00 | 16 | -0.8 (AP) / +0.1 (noAP) |
| 20191227 | 08:00 – 11:00 | 15 | -0.2 (AP) / +0.3 (noAP) |
| 20191228 | 08:00 – 11:00 | 29 | -0.2 (AP) / +0.1 (noAP) |
| 20200102 | 03:00 – 05:00 | 70 | -0.4 |

Table 2: Ground radar – OceanPOL calibration difference estimates for all comparisons of this study. A mean
calibration difference for radars 63 and 77 that includes all dates and time spans is also provided. For radars 15, 16,
and 29, two estimates are provided, with no test on minimum height (AP) or with a minimum height of 2 km for the
comparisons (noAP), in an attempt to remove residual anomalous propagation artefacts observed for these radars.


**Figures**

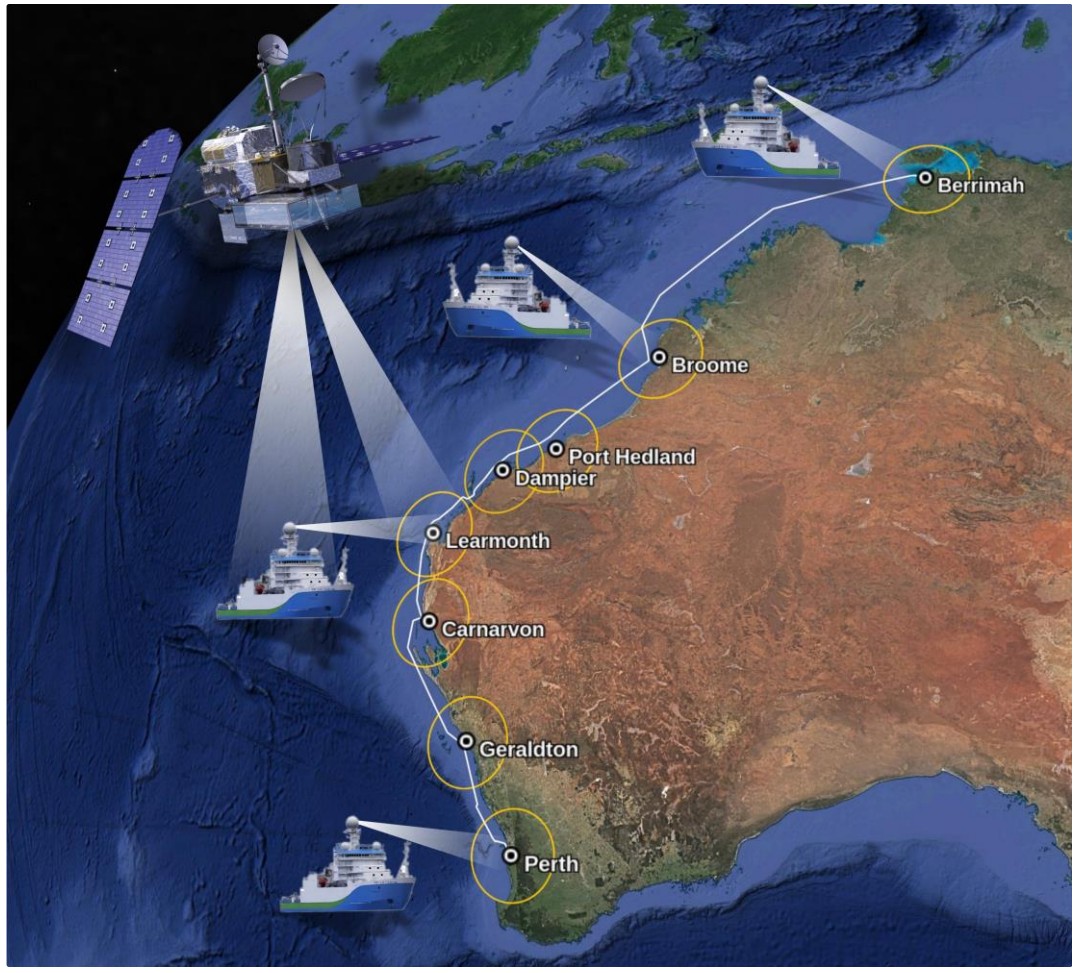


**Figure 1: The concept of this study. Ship-based OceanPOL radar and ground-based radars are calibrated independently**
**using the GPM Ku-band spaceborne radar, then all ground radars are compared with OceanPOL during the ORCA**
**voyage as RV Investigator sails south. The 150 km radius of each radar is shown by a yellow circle and the ship track is**
**shown using a white line. © 2021 Google Earth; Map Data: SIO, NOAA, U.S. Navy, NGA, GEBCO; Map Image:**
**Landsat/Copernicus.**

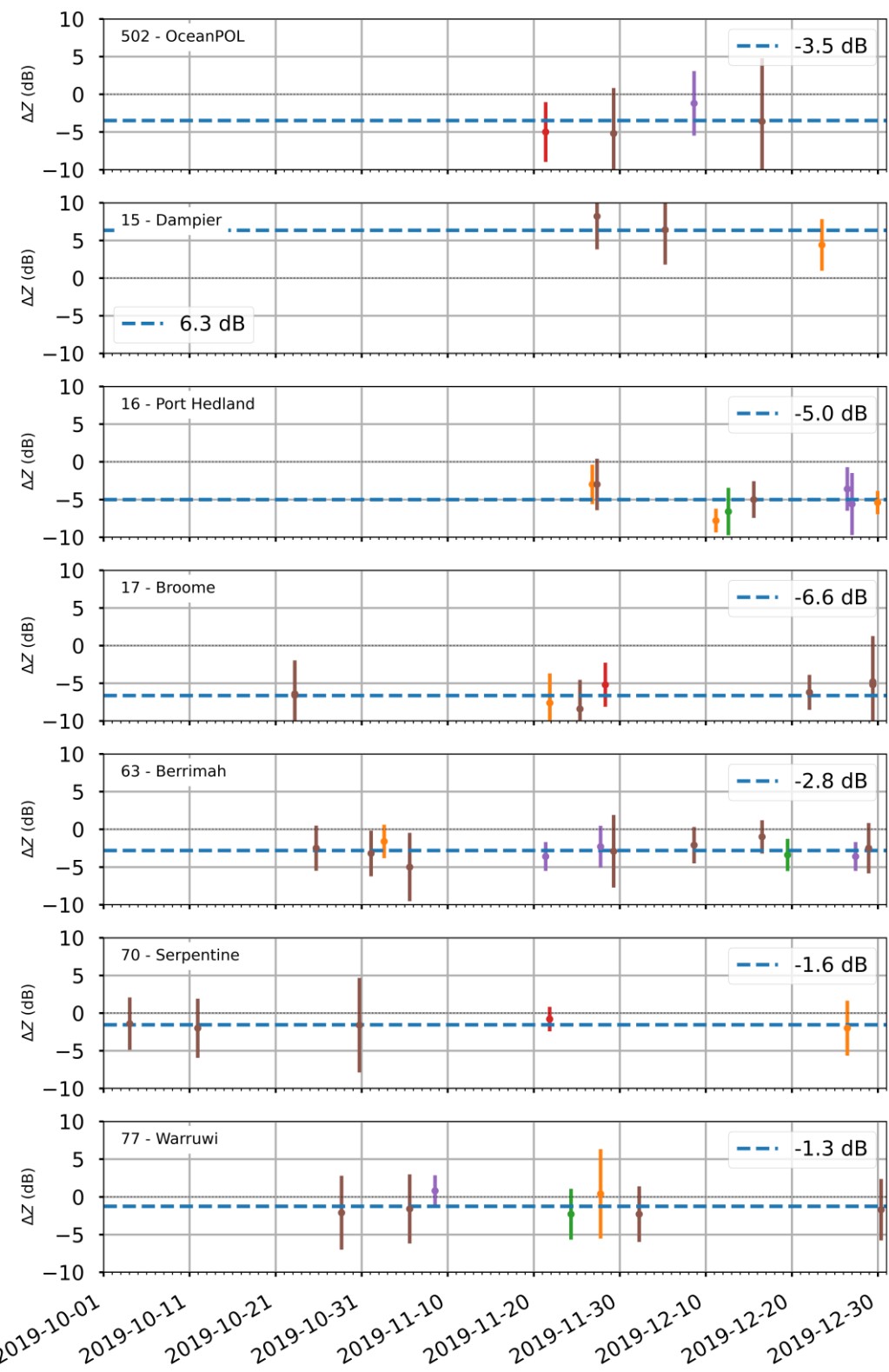


**Figure 2: Individual calibration error estimates from the GPM comparisons, for all radars used in this study. The standard deviation of the PDF of reflectivity difference is also shown for each estimate as an error bar. The mean value over the whole period is displayed as a dashed line for each radar, and the value is reported on the upper-right of each panel. Note that a negative value mean that the radar is under-calibrated (radar – GPM). The colour of each overpass point is the number of matched volumes: less than 20 (blue), 20 to 60 (orange), 60 to 100 (green), 100 to 150 (red), 150 to 200 (purple) or more than 250 (brown).**

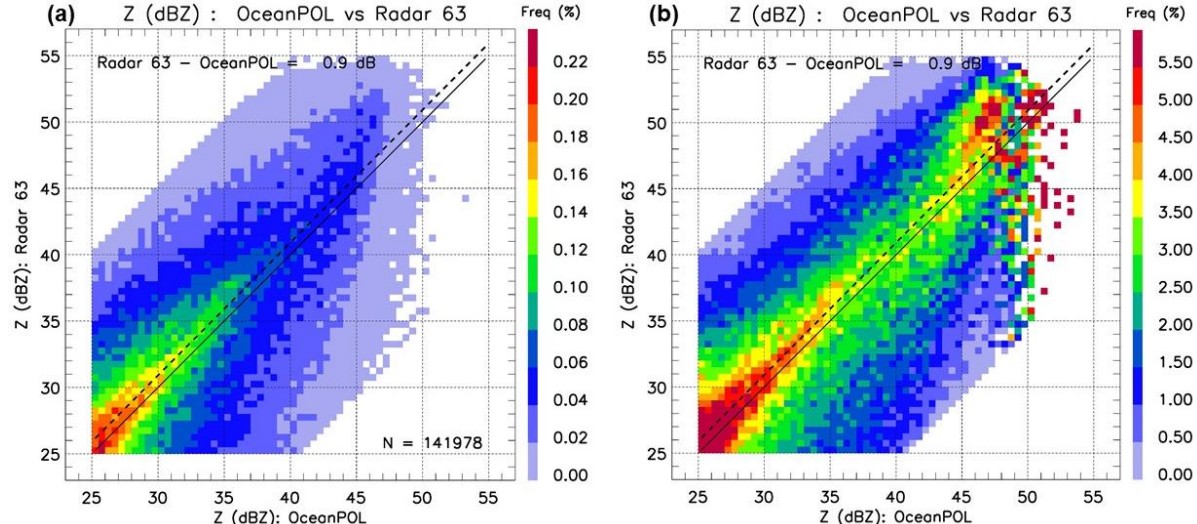

**Figure 3: Illustration of 2D joint frequency histograms of reflectivity used to compare quantitatively the OceanPOL radar (x-axis) and any of the ground-based radar (y-axis), here for the Berrimah radar (63) for one day (21 November 2019) of the YMCA experiment. For each plot, the 1:1 line is drawn as a solid line, and the calibration difference estimate is written and shown as a dashed line. The colours show the frequency of points falling in each reflectivity pixel 0.5 dB in resolution of the 2D joint histograms, either expressed as the % of the total number of points (panel a) or as a % of the sum of points for each value of OceanPOL reflectivity (i.e., sum of all points along the y-axis at each constant value of the x-axis). The number of samples N for this case is 141978 (see panel a).**

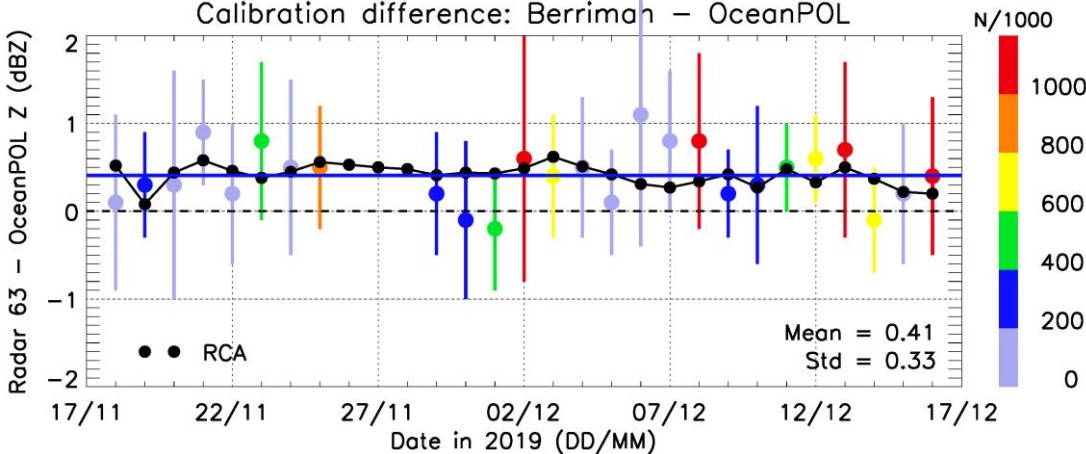

**Figure 4: Time series of calibration differences between OceanPOL and radar 63 (Berrimah) during the YMCA experiment. Each coloured point is a daily estimate of calibration difference. The colour of the point is the number of points for each comparison, and the coloured error bar is the standard deviation of hourly calibration difference estimates for that day (see text and Fig. 6 for more details). The solid blue line is the mean value obtained from all these daily estimates (0.4 dB). The overall mean and standard deviation of the daily calibration difference over the period of observations are also written on the lower-right side of the figure. The black dashed line is the zero line. The black points are the daily outputs of the RCA values, with the mean RCA value over the period subtracted and the mean value of calibration difference added, so that the time series is centred on the mean calibration difference value.**

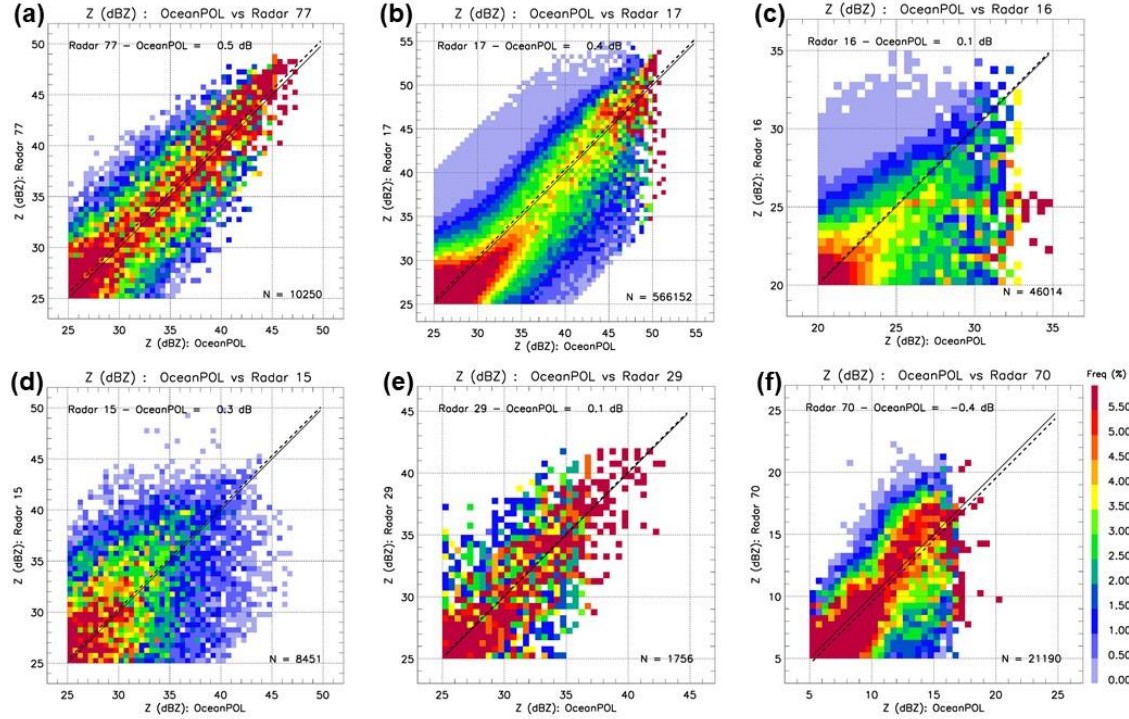

**Figure 5: 2D joint histograms of reflectivity as in Figure 3b but for radars (a) 77, (b) 17, (c) 16, (d) 15, (e) 29, and (f) 70.**
**Values of calibration differences are also reported in Table 2. The number of samples N is also given in each panel.**

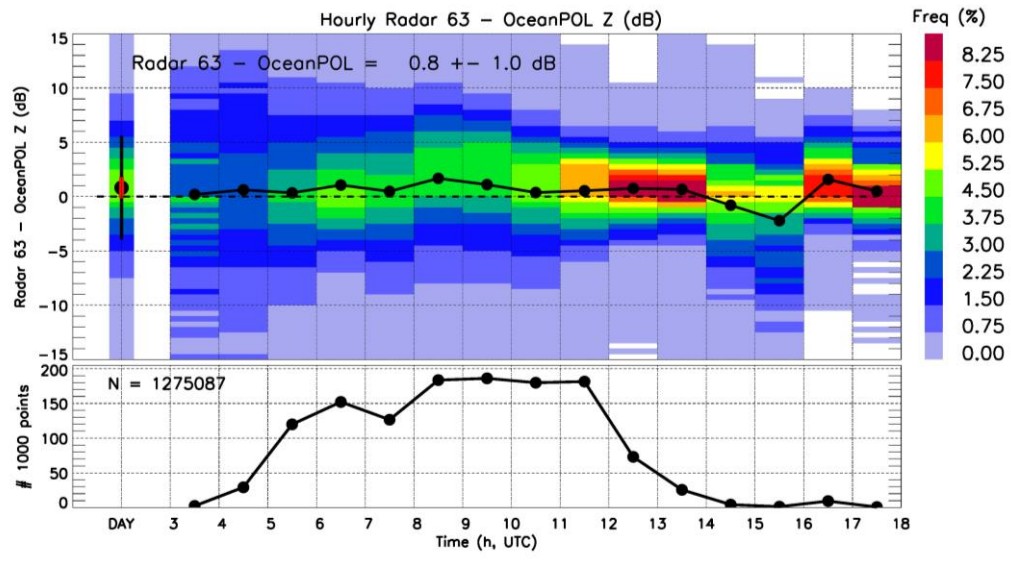


**Figure 6: Hourly analysis of calibration differences between Berrimah (radar 63) and OceanPOL for a selected day**
**(08/12/2019). The upper panel shows each hourly calibration estimate as a black dot, as well as the full frequency**
**distribution of differences within each hour (colours). The first column of the upper-panel shows the daily summary,**
**including the mean value (black dot, value is also written), the frequency distribution of calibration differences (colours),**
**the standard deviation of the difference using the N collocated samples (black error bar), and the standard deviation of**
**the hourly estimates of calibration differences for that day (red error bar, value is also written). Lower panel shows the**
**number of samples in each hour (note y axis is the number of points divided by 1000) and the total number of samples N**
**is also provided.**