# Peer review of "Three-way Calibration Checks Using Ground-Based, Ship-Based and Spaceborne Radars"

_Atmospheric Measurement Techniques, 2021_

## Referee Comment (RC2)

Review of the manuscript to AMT entitled

**"Three-way Calibration Checks Using Ground-1 Based, Ship-Based**

**2 and Spaceborne Radars"**

by  A. Protat , V. Louf , J. Soderholm, J. Brook, W. Ponson

**General comment:**

The authors evaluate and assess the accuracy achieved of the recently developed radar calibration framework used to monitor the calibration accuracy of all operational radars of the Australian weather radar network in real-time. The technique is based on the comparison with spaceborne Ku-band radar observations from GPM applying a Volume Matching Method (VMM). After an additionally available ship radar (OceanPOL) and the radars from the network have been calibrated separately by comparison with the space-born measurements, measurements of all radars of the Australian network are compared to the ones of OceanPOL. These more accurate ship – ground radar comparisons are considered as an indirect evaluation of the GPM validation technique and exploited to demonstrate the value of using such GPM data as a single source of reference for the calibration of a whole national network. Indeed, for all seven radars the calibration difference with the ship radar lies within ± 0.5 dB.

Intercomparisons of gridded radar observations also revealed the potential to estimate calibration differences between radars with overlapping coverage to within about 0.3 dB at daily time scale and about 1 dB at hourly time scale, which can be exploited for additional calibration monitoring.

The accuracy and value of the outlined calibration strategy is of interest for the community. Furthermore, my list of edits and suggestions provided below includes nothing severe and therefore I suggest this manuscript for publication after their consideration. E.g. at several places the authors refer the reader to upcoming 'later' explanations without being precise. In case references to later subsections are required more often, restructuring of the manuscript may also be an option. At one or two places the introduction of subsections would make the structure of the text more transparent and for some aspects I am also missing some more detailed explanations (see 'major' points, even though they are not really major, but I distinguish them from pure formulation issues).

**Major points:**

Line 13: Maybe the advantage of using a ship-based radar should be shortly indicated here?

Lines 16/17: What about the range of differences before the calibration? Please see also my comment regarding line 165.

Line 29: The pointing accuracy is not mentioned again in this manuscript, right?

Lines 38/39: Are there references demonstrating/documenting the accuracy of the GPM radar? You also say in Line „…whose calibration is very accurately tracked by NASA." Can you be more precise here?

Line 50: Reads a bit weird that the advantages of using the ship radar will be discussed ‚later' (no precise statement), followed by the precise structure of the article in the following lines (section 2, section 3, section 4 contains this and that).

Lines 79, „more accurate source of reference": For the third time the authors indicate here the special role /higher accuracy of the ship radar without explanation. I would suggest to unravel the secret earlier. I was wondering already while reading the abstract, why the ship radar is a reference. If I understand correctly, the solution is provided in lines 192ff and I would suggest to summarize this idea also in the abstract.

Line 91: What kind of additional quality control is done?

Line 123: Just for curiosity: Quite often ring structures are generated with gridding/compositing of radar data. Do you encounter similar problems? If yes, this could also impact the comparison.

Line 129ff: Not really clear to me what is done. The radius of incluence is only applied to the same elevation but how do you decide whether an adjacent elevation is included?

Line 139: What about corrections for the ship movement? I thought such kind of things introduce different uncertainties for ship radars, but the authors only mention the superiority of OceanPOL.

Line 170, „discussed later and shown as black dots in Fig. 4)": Such references to later paragraphs should be avoided if possible, or at least be more precise. When/where instead of ‚later'? Or restructuring the manuscript should be considered in case too many references to later paragraphes are needed.

Line 188, frequency conversion: Should be mentioned how this is done/taken into account.

Line 188, most problematic in the melting layer: The melting layer is not excluded from the comparison? I suggest to be more precise how the comparison is performed.

Line 214, „We will get back to that point shortly.": Again, refering to the unprecise future is not optimal.

Line 247: Maybe nicer, more structured, to introduce 2 subsections 3.1 and 3.2, with subsection 3.2 starting here dealing with the day-to-day variability.

Line 254: Again refering the reader to the unkown future „which will be discussed in more detail later." Please avoid.

**Minor points:**

Lines 81 and 83 contradict each other: Table shows all radars used in this study including different frequencies, but then it says this study uses only the C-band radars.

Line 90: Did you ever applied consistency of polarimetric variables for calibration of OceanPOL and checked the agreement with your calibration based on the GPM measurements?

Line 91, Version 5 of the GPM 2AKu product: A reference would be nice here.

Line 137/138: Are you using stratiform and convective events for the comparison?

Line 165, „All calibration results are summarized in Fig. 2.": The calibration results should be mentioned in the text, not only written in the panels of the Fig.

Line 168/169: The older estimate also includes RCA checks or just for radar 63?

Line 173, „Looking at the time series of GPM calibration estimates for other radars than 63 …": Why not refering now to radars 63 AND 29?

Lines 173ff: So, at the end only radar 16 shows variations? Why not directly writing that instead of starting with 29, then adding 63 and finally the others? Would be easier to read.

Line 182, ‚In a perfect world'…: I am not a native speaker, but sounds more like colloquial language to me.

Figure 3 caption: Here I suggest to write that the comparison with radar 63 is shown, but in the text you can write that the overall strategy for the OceanPOL comparison with any radar is illustrated using radar 63 in Fig. 3. I also suggest to rewrite the explanation of what is shown in panel b. Maybe something like „…a percentage of all OceanPOL reflectivity values in a resolved 0.5dB bin". And I suggest to write „The number of samples N is 141978 (see panel a)."

Line 200: Instead of „as on the left panel of Fig. 3)" -> Fig. 3a

Line 202: Better „Comparison of …provides a better….and allows the detection…"

Line 206: Better „reflectivities less than 35 dBZ mostly contributed…"

Line 210: Delete ‚and'

Line 215, „When including all days of observations for radars 63 and 77": Is there a need to emphasize this? For the other radars not all measurements available are used?

Lines 217: ‚See' Fig. 4 for ….

Line 218, „The next best operational radar is radar 70 (Perth).": Please reformulate.

Line 289: …derived from all hourly not from all daily estimates, right?

Lines 304/305: „A major advantage of using a single source of reference is that all radars of the network are calibrated in the same way." This is also the case when other methods are used. If you choose to use the consistency method for your network, you also use the same method for your entire network. Not sure what you want to express here.

Lines 305-308: I guess this sentence can be better formulated. Bit hard to read.

Lines 350ff: For several publications the DOIs are provided, for others not.

---

## Author Comment (AC1)

Responses to reviewers

Reviewer 1:

Thanks for this extensive review, which we believe helped improve the quality of our manuscript. Below we address all comments (in red).

Monitoring and maintaining the calibration of a weather radar network is an important task. Especially for dualpol radars multiple complementing monitoring sources are needed to assess the calibration of a radar. This work considers one particular source to monitor the calibration of ground based weather radars. Using the NASA GPM, it is shown that there is systematic negative bias of the surface bias for all radars, assuming that the GPM is considered as a references. Among the weather radars, the bias to the GPM data varies between -6.6 and -1.3 dB which is significant, but in part appears to be attributed to the differences in hardware and the age of the systems (without actual prove). But there seems to be no attempt trying to explain the large biases. On the other hand, dedicated ship born radar measurements compared to selected radars illustrate a consistence of the measurement within +/-1 dB, which includes the radar Broome. Using shipborne radars to assess the calibration of operational weather radars is unique.

For the radar Broome the authors find a bias of -6.6 dB compared to GPM data. The authors do not attempt to find the source for those biases, and there is no plan laid out on the next steps. The relative consistence of the shipborne radar data and the continental radars compared to the large variability found between the GPM radar data and the ship radar /radar network suggest that a much more thorough investigation is needed before the satellite data can be one source of the calibration monitoring of the Australian weather radar network. In the conclusion (l.303) you state that you want to get "insights into the accuracy" of space borne radar obs "to calibrate national operational radar network". I would argue that you missed this goal. In principle you compare reflectivities doing a careful designed radar/radar comparision without going into the details explaining obvioue differences.

It may come as a surprise to Reviewer 1, but it is actually quite common to have operational radars miscalibrated by as much as 7 dB (we have some remote radars that are miscalibrated by that much at the moment), either long-term or intermittently when something suddenly fails and remains unnoticed for a period of time. There are many underlying reasons for that, which I am not going to get fully into, but here are the main ones :

- radar engineers only do "internal calibration", which means that they measure the gains and losses of blocks of components inside the transmitting and receiving chains. This is not a complete end-to-end calibration.
- What is perhaps more surprising is that the devices they use to make those measurements also need calibration, and I can give you a very relevant anecdote that in the Northern Territory, during wet season 2018-2019, all radars were systematically miscalibrated by about 4 dB, but quite consistently across all radars. That was due to a problem with the calibration of the measurement devices they used ! An internal program has been developed since then to do regular calibration of these devices and swap between regions.
- Probably the most important one in the context of Reviewer1's comment is that radar maintenance, which includes these calibration measurements, happens every 6 months at best. Until we developed this radar monitoring tool, nothing could tell you that something was wrong with the radar calibration. Any problem with the transmitter or any part of the radar would remain undetected. For remote radars such as Broome, it is very common to skip

a 6-monthly calibration check as well. I don't think other met services around the world, even in the US, do things very differently.

Hopefully with these simple facts, we have convinced Reviewer 1 that with 6-monthly calibrations anything can happen in between two checks, including a significant failure of the transmitter explaining a 6 dB miscalibration.

If you need more convincing that radar calibration is indeed a dark art, please take a look at the Kollias et al. (2020) paper. Here you have one of the most advanced radar programs in the world from the US DoE Atmospheric Radiation Measurement (ARM) program. Please take a look at the miscalibrations reported after radar engineers have claimed their radars were perfectly calibrated.

*Kollias, P., B. Puigdomènech Treserras, and A. Protat, 2019: Calibration of the 2007-2017 record of ARM Cloud Radar Observations using CloudSat. Atmos. Meas. Tech., **12**, 4949–4964.*

There is another point I want to make to reply to this specific criticism from Reviewer 1:
" On the other hand, dedicated ship born radar measurements compared to selected radars illustrate a consistence of the measurement within +/-1 dB, which includes the radar Broome".

Unfortunately, it appears that the reviewer has missed an important detail of the exercise, so we have revised the text to try and make the point across more clearly. In short, we have calibrated all the radars (including the shipborne radar) using GPM. Using a single source of reference (GPM), no matter what the quality of the GPM calibration is, we expect that then when we compare ground and ship radars they should be consistent. That is indeed what we found (+-1dB). Residual differences are interpreted as uncertainties of the GPM comparison technique. So, I absolutely don't think we have missed the goal of the paper.

These points have been clarified in the introduction to the paper as follows: "" The concept of this study is presented in Fig. 1. GPM observations are first used to calibrate both the ship-based radar and all the operational ground-based radars along the coast independently. The ship-based radar observations calibrated using GPM are then individually compared with those from each ground-based radar as the ship sails close to them. Since all radars (including OceanPOL) have been calibrated using GPM, the differences between ship-based and ground-based observations can be used as an error estimate of the GPM calibration technique, with some contribution from errors due to the ship-ground radar comparisons themselves. However, these errors are expected to be much lower than those arising from the GPM / ground radar comparisons. Indeed, the advantage of using a ship-based radar relative to a spaceborne radar is that many of the error sources in ground-based / satellite radar comparisons are reduced to a minimum. "

A more thorough literature survey is missing. This should include references on how other met services monitor the calibration of their radars. I would expect at least a brief summary on how the BOM maintains and calibrate the operational radar network, , how the calibration is done on the ships. Since this is a paper that deals with calibration, this is essential to me.

Very good point. We have added the following text to explain how calibration is checked operationally at the Bureau :
"The internal calibration accuracy of these operational radars is checked six-monthly by BoM radar engineers as part of their routine maintenance. The calibration check only includes measurements of gains and losses at different check points of the transmission and reception chains. No end-to-end calibration using external targets is ever performed. Special visits to sites are organized when a radar is down or when complaints are issued by the public about radar data quality"

The following text has been added for OceanPOL:
"The same calibration procedure as that employed by BoM is used for OceanPOL (internal measurements of gains and losses, no end-to-end calibration)."
In terms of literature review, we have looked into material explaining how met services do this, and I must report that I could not find anything in the peer-reviewed literature, except for the specific case of the Korean dual-polarization radar network. I then contacted colleagues at NOAA and NSSL to get more insights into what is being done for the NEXRAD network, and the answer was that there is no readily available literature on this. Informally, I've been told that the Radars Operations Centre in the US is currently reviewing the way they monitor calibration because there have been several reports indicating that the calibration of some NEXRAD radars is very questionable.

A more general comment: the term calibration is sometimes used in a very loose way. When doing calibration a normed reference is used to determine the calibration. I wouldn't consider the GPM as a "normed reference" (see e.g. wording in l.39) since it is also an remote sensing instrument which has to be calibrated. So I suggest to be more careful with the wording throughout the manuscript.

Yes, we do agree with both aspects of this comment.
- The term calibration can formally only be used to refer to the engineering calibration against a reference source or through end-to-end measurements of gains and losses. We note however that it is customary to use this term to refer to any attempt to get closer to the true calibration as we do here.
- Completely agree that GPM is not a normed reference. However, the effort that goes into absolute GPM radar calibration and calibration changes is a lot higher than what is being done with ground-based radars. Procedures to monitor all gains and losses is lot more thorough. Comparisons with ground references is done regularly.

This very good point has now been disclosed in the text explicitly:
" In what follows, we will use the term "calibration" to refer to calibration differences between ground or ship-based radars and the GPM radar taken as the "reference". However, it must be noted that reflectivities measured by the GPM radar are not a normed reference, which implies that our use of the term "calibration" is strictly not correct. "
Following this comment, we also decided to add more information about how thorough the GPM calibration is, which reads as follows:
"As discussed previously, the GPM Ku-band reflectivities are considered as the reference for the calibration of all radars in this study. The GPM radar calibration procedure, described in detail in Masaki et al. (2020) inherited from years of calibration work undertaken as part of the previous satellite radar mission, the Tropical Rainfall Measurement Mission (TRMM). This calibration comprises an internal calibration (monitoring closely the gains and losses of each component of the radar) and an external calibration procedure using a ground-based calibrator and sea surface of well-known backscatter. Importantly, the GPM mission also benefits from extensive field experiments undertaken as part of the Ground Validation program, including in-situ ground and aircraft validation of the products of the GPM mission. By comparing different approaches for the GPM Ku-band radar calibration, Masaki et al. (2020) demonstrated that the accuracy of the radar was well within the ±1 dB requirement."

To conclude, this is in principle is an important investigation, which falls short on assessing sources for calibration errors and the observed relative differences between the sensors. Without such an assessement, the results remain inconclusive.

We hope we have now convinced that this general comment is unwarranted.

Some more specific comments:

How do you do the frequency correction from Ka -> C band? Or is there already a C-Band product you can use?

Thanks for picking up that we forgot to mention what we used for this. We added the following text at the end of section 2.2:
"To compare GPM Ku-band radar with C-band radars in this study, all GPM Ku-band reflectivities have been converted to their equivalent C-band reflectivities using Eq. 5 in L19."

l 95: what is "dark art" about calibration???

That was a tongue-in-cheek title to make the point that this is not a simple task. We changed the title to a more formal " The S$^3$CAR radar calibration framework "

l 115:  Please include the solar monitoring  results for the Berrimah and Geraldton radar. That would be helpful to understand possible error sources. I miss the Geraldton  radar in the results. Why is it missing?  (it is given in Figure 1)

As we clearly explained lines 76-77, there was no precipitation when we were sailing near Geraldton and Carnarvon, so Geraldton is one of the two radars for which we do not have collocated samples with the shipborne radar. That is why 1) you did not find results about this radar, and 2) there is no point showing the solar monitoring results for this radar.
Regarding Berrimah, the RCA results are shown in Fig.4. The accuracy of the RCA to track calibration change is higher than what you can achieve with the conversion of solar power to an equivalent radar reflectivity. In addition, from the RCA results, there appears to be no significant calibration change over that period, therefore no error sources explaining a change in calibration to understand. As a result, we are not sure what the solar calibration results would bring in the discussion.

l.116: raw reflectivities: I assume you mean unfiltered data, no clutter correction applied, no range averaging? Please state clearly what you mean with "raw"

Yes, sorry that is an internal term we always use to describe reflectivities directly as measured with no corrections applied to them, no filtering, no noise or ground clutter removal etc … Unfiltered data is also not a comprehensive way of describing this (we don't just do filtering on raw radar data). We have decided to call these "unprocessed" to highlight that there has been no processing done on them, we hope that's a suitable term. We have also added this to make the point clearly :" ("unprocessed" refers to data still containing noise and all typical radar signal contaminations, including ground clutter and sun spikes used in our calibration techniques)"

l 121 ff: there is no need to separate in HM type depending the height of the radar bin? Why not?

Sorry, we don't understand what this question is about. Line 121 is the title to section 2.4 so it's difficult to understand what is being referred to. Also what is HM?

l. 404: Table 2: please include the names of the radars instead of the numbers… makes it easier to read.

Agreed. Done.

L 417: Figure 2:  no data for Learmonth? why showing this graph?

Yes, no point, we fully agree. This panel has been removed.

L 426: Figure 3:   from the caption, the difference in (a) and (b) is not explained. Please describe briefly, as the captions should be self explaining. Or refer to the text.
Sorry, but this is already explained in the caption: " The colours show the frequency of points falling in each pixel of the 2D joint histograms, either expressed as the % of the total number of points in (a) or as a % of the sum of points for each value of OceanPOL reflectivity (i.e., sum of all points along the y-axis at each constant value of the x-axis) in (b)."

Dampier with 6.3 dB  bias (Fig 2) : did you check the calibration procedure? How often are radars calibrated in the network? Should be mentioned somewhere. A 6.3 dB bias  ia dramatic.
Fully agree that such high value is a real problem. Even more problematic is the fact that such calibration error is not for a short period of time but was present for at least a month. As we explained in our general response, and now in the manuscript, radar calibration checks are scheduled every 6 months only, and for remote radars like this one, sometimes a full year. Until we developed S$^3$CAR, there was no way to detect issues like this. We are now discussing internally how to leverage from this new information to uplift our internal calibration schedule. The idea is that they could go less often if we don't see issues, but they would have to go earlier if an issue was detected.

L 271 "natural variability of the calibration figure": I  would suggest to reserve the term calibration for a "real calibration" where you compare against a normed reference. Here you consider a relative adjustment (so far you quantify the difference, but you seem to suggest the satellite could be taken as the truth).

Yes, our assumption in this work is that the GPM radar is well calibrated and is therefore our "reference". As discussed in our general response, we agree that formally we should not use the word "calibration", so we now disclose this improper use of the term calibration in the text.

You only discuss the relative differences between satellite and surface/ship based observations. A bias of 3 dB for your ship based radar compared to GPM is significant. I'm a bit surprised that you don't try assess the possible source of the bias. I assume that this instrument has more staff to do a more thorough investigation on the technical aspects of calibration, to really pin down (or rule our) the origin of the bias, checking all relevant elements of you radar hardware, and use the sun as reference to verify your receiver calibration. This should include also a check of the GPM data, perhaps using other data sources like disdrometer measurements (if available)

See general response on this point. Engineering calibration is not an absolute calibration, it involves measurements of gains and losses of components inside the radar but does not include everything or not perfectly either. If you talk to any radar engineer, they are generally not surprised if you tell them that their radar calibration is out by 3 dB. The ship-based system is monitored by a single engineer, who's also maintaining a lot of other instrumentation on the ship. With support from my team, absolute sphere calibration procedures are being put in place by the CSIRO radar engineer as we speak, to do exactly what you suggest. We should soon be able to better control and maintain an accurate calibration for this shipborne radar.

I wonder how the GPM products compare to rain gauge estimates in Australia? Any hint? Would be worthwhile in the literature survey.

Rain gauge comparisons are probably the least accurate way of calibrating a spaceborne radar I could think of. Differences in sampled volumes, issues with ground clutter contamination for the satellite radar, differences in heights, and inaccurate conversion of reflectivity to rainfall or conversion from rain gauge rainfall to reflectivity using scattering calculations … I don't recommend this at all to calibrate a radar, especially spaceborne. It's not even part of our S$^3$CAR suite of tools for ground-based radars for those reasons, although we have a well-developed rain gauge network around some of our radars.

Reviewer 2:

The accuracy and value of the outlined calibration strategy is of interest for the community. Furthermore, my list of edits and suggestions provided below includes nothing severe and therefore I suggest this manuscript for publication after their consideration. E.g. at several places the authors refer the reader to upcoming 'later' explanations without being precise. In case references to later subsections are required more often, restructuring of the manuscript may also be an option. At one or two places the introduction of subsections would make the structure of the text more transparent and for some aspects I am also missing some more detailed explanations (see 'major' points, even though they are not really major, but I distinguish them from pure formulation issues).

Thanks for a very constructive and positive review. We have addressed all these comments. Please find below our individual responses to each of them (in red).

Major points:

Line 13: Maybe the advantage of using a ship-based radar should be shortly indicated here?
Good point. We have added this : " The calibration difference between the OceanPOL radar *(used as a moving reference for this second step* "

Lines 16/17: What about the range of differences before the calibration? Please see also my comment regarding line 165.
The range of differences before the calibration is somewhat irrelevant to the point we are trying to make in this study. However, this comment made us realize that we had not really described the operational maintenance routine from the Bureau, which largely explains these mis-calibrations. We have added the following text when describing the different radars of this study:
"The calibration accuracy of these operational radars is checked six-monthly by BoM radar engineers as part of their routine maintenance. The calibration check only includes measurements of gains and losses at different check points of the transmission and reception chains. No end-to-end calibration using external targets is ever performed. Special visits to sites are organized when a radar is down or when complaints are issued by the public about radar data quality"
The following text has also been added for OceanPOL:
"The same calibration procedure as that employed by BoM is used for OceanPOL (internal measurements of gains and losses, no end-to-end calibration)."

Line 29: The pointing accuracy is not mentioned again in this manuscript, right?
Yes, that's correct. As this could be distracting, we have removed the reference to this other aspect of the monitoring we do with our S$^3$CAR tool.

Lines 38/39: Are there references demonstrating/documenting the accuracy of the GPM radar? You also say in Line „…whose calibration is very accurately tracked by NASA." Can you be more precise here?
Another good point. Since we rely on GPM to provide a reference calibration, it does make sense to explain this in more detail in our paper. Wait for Walt's response to my email.

Line 50: Reads a bit weird that the advantages of using the ship radar will be discussed 'later' (no precise statement), followed by the precise structure of the article in the following lines (section 2, section 3, section 4 contains this and that).

Agreed. We have removed this reference to a discussion that will happen later. And we have started a new paragraph to make a clear separation between the motivation of the study and the description of the structure of the paper.

Lines 79, „more accurate source of reference": For the third time the authors indicate here the special role /higher accuracy of the ship radar without explanation. I would suggest to unravel the secret earlier. I was wondering already while reading the abstract, why the ship radar is a reference. If I understand correctly, the solution is provided in lines 192ff and I would suggest to summarize this idea also in the abstract.

Yes that is a very valid point too. To address this comment (and some of the previous ones), we have brought forward (in the introduction) more description about this to be clear upfront about why using a shipborne radar sheds light on the GPM calibration technique error estimate:

" The concept of this study is presented in Fig. 1. GPM observations are first used to calibrate both the ship-based radar and all the operational ground-based radars along the coast independently. The ship-based radar observations calibrated using GPM are then individually compared with those from each ground-based radar as the ship sails close to them. Since all radars (including OceanPOL) have been calibrated using GPM, the differences between ship-based and ground-based observations can be used as an error estimate of the GPM calibration technique, with some contribution from errors due to the ship-ground radar comparisons themselves. However, these errors are expected to be much lower than those arising from the GPM / ground radar comparisons. Indeed, the advantage of using a ship-based radar relative to a spaceborne radar is that many of the error sources in ground-based / satellite radar comparisons are reduced to a minimum. "

Line 91: What kind of additional quality control is done?

The point here is that although OceanPOL is a dual-polarization radar we are not using this capability anywhere except for the quality control. Our dual-pol quality control, just like others, includes the detection and removal of non-meteorological echoes, which is the main advantage of dual-polarization for quality control. Implicitly, it means that the quality control of OceanPOL is superior to that of single-pol radars.

Line 123: Just for curiosity: Quite often ring structures are generated with gridding/compositing of radar data. Do you encounter similar problems? If yes, this could also impact the comparison.

Yes, those rings are potentially a major artefact in radar gridding techniques. As it turns out, we are currently finalizing a paper discussing this in detail and how to mitigate them in modern gridding techniques. They are generated by the use of a radius of influence in the vertical, which we avoid doing in our own gridding technique by using linear interpolation in the vertical instead (see also response to your next question). The impact of linear interpolation is that you generate gaps instead of filling in artificially with data that are dragged by the radius of influence from neighbouring regions. For our study, it is definitely advantageous to leave unobserved data s gaps rather than trying to fill them in.

Line 129ff: Not really clear to me what is done. The radius of incluence is only applied to the same elevation but how do you decide whether an adjacent elevation is included?

Adjacent elevations below and above each grid are used to produce a linear interpolation onto the grid location. This was not well explained in the text, so we have added this in the new version :
"a linear interpolation in the vertical using data from the elevations below and above each grid"

Line 139: What about corrections for the ship movement? I thought such kind of things introduce different uncertainties for ship radars, but the authors only mention the superiority of OceanPOL.

Thanks for picking that up. We had completely forgotten to talk about this in our description of the OceanPOL radar, although it is a critical aspect of operating a radar from a ship. We have added the following paragraph in the section describing OceanPOL:

" A critical aspect of operating a radar on a research vessel is the need to compensate for ship motions and velocity in real-time. To do so, the OceanPOL antenna control system ingests the real-time inertial motion unit data from the ship at 10 Hz and steers the radar beam in real-time in the requested azimuth and elevation direction. The accuracy of this stabilization has been found to always produce a pointing accuracy better than 0.1°, even in harsh sea conditions. Doppler measurements are automatically corrected in real-time for the Doppler component induced by ship velocity components. Dual-polarization moments are also corrected using the statistical corrections proposed in Thurai et al. (2014). "

Line 170, „discussed later and shown as black dots in Fig. 4)": Such references to later paragraphs should be avoided if possible, or at least be more precise. When/where instead of ‚later'? Or restructuring the manuscript should be considered in case too many references to later paragraphes are needed.
Removed.

Line 188, frequency conversion: Should be mentioned how this is done/taken into account.
Thanks for picking up that we forgot to mention this. We added the following text at the end of section 2.2:
"To compare GPM Ku-band radar with C-band radars in this study, all GPM Ku-band reflectivities have been converted to their equivalent C-band reflectivities using Eq. 5 in L19."

Line 188, most problematic in the melting layer: The melting layer is not excluded from the comparison? I suggest to be more precise how the comparison is performed.
When writing this paper, we considered but discarded that option of describing more fully all the details of the GPM volume matching technique because this has been fully documented in Warren et al. (2018) and Louf et al. (2019). Instead, we clearly mention that we have used exactly the L19 technique. However, we do agree that more details were required to document what has been done to mitigate the errors sources. To address this comment, we have added the following paragraph in section 2.3:
" The operational implementation of the GPM calibration technique closely follows the description given in W18. Satellite and ground-based radar observations are first matched to a common volume. We require at least a minimum of 10 satellite profiles within the ground radar domain to select and process a satellite overpass. The melting layer is detected by the operational GPM algorithms and excluded from the matched volumes due to uncertainties in frequency conversions for melting hydrometeors. Matched volumes in both liquid and ice phases are retained (like in W18). Non-uniform beam filling effects of the matched volumes are mitigated by only selecting volumes that are 95% filled. Minimum and maximum ground-based reflectivity thresholds of 21 and 36 dBZ are used in the analysis of matched volumes to mitigate the impacts of sensitivity differences (see Fig. 8d in W18 for more details on this) and attenuation correction errors, respectively."

Line 214, „We will get back to that point shortly.": Again, refering to the unprecise future is not optimal.
Removed (unnecessary).

Line 247: Maybe nicer, more structured, to introduce 2 subsections 3.1 and 3.2, with subsection 3.2 starting here dealing with the day-to-day variability.

We considered that option of subdividing the results section and followe that recommendation from Reviewer 2, but we opted for three subsections, one grouping the first set of results under "Accuracy of the GPM VMM technique", one describing the day-today variability and one discussing the hourly results. This results in one subsection being a lot longer (the first one) and the two others being merely a paragraph or two. But we still think it helps navigate in the results section a lot better, so thanks for this suggestion.

Line 254: Again refering the reader to the unkown future „which will be discussed in more detail later." Please avoid.

Removed (unnecessary).

Minor points:

Lines 81 and 83 contradict each other: Table shows all radars used in this study including different frequencies, but then it says this study uses only the C-band radars.

Yes, there was something weird about this sentence, as the opening statement did not apply to radar frequency. We have rephrased as follows : " The Australian radar network comprises a large variety of radars from different generations, frequencies (although ground-based radars in this study are all C-band radars, other parts of the country are covered by S-band radars), …"

Line 90: Did you ever applied consistency of polarimetric variables for calibration of OceanPOL and checked the agreement with your calibration based on the GPM measurements?

Yes, we have done such preliminary work over the Southern Ocean, but not in this particular region. That is definitely something we want to investigate in the near future, as soon as we have collected relevant observations. What we have shown in Louf et al. (2019) is that the regional variability of this self-consistency relationship is large. As a result, we cannot use our Southern Ocean relationship to do this. The reason why we did not derive a tropical relationship in the present study is that there has been very little rain exactly over the ship during the YMC and ORCA campaigns, which precludes any statistical analysis of our disdrometer observations to produce a robust self-consistency relationship for this region.

Line 91, Version 5 of the GPM 2AKu product: A reference would be nice here.

I could only find a reference for the validation of the Version 5 products, and a link to the products themselves on the PMM database. So I added that information.

Line 137/138: Are you using stratiform and convective events for the comparison?

We have now explained following your comments what reflectivities are retained for the analysis of GPM / ground radar comparisons. Regarding ground-ship comparisons, we retain all data without discriminating convective from stratiform.

Line 165, „All calibration results are summarized in Fig. 2.": The calibration results should be mentioned in the text, not only written in the panels of the Fig.

As discussed previously, the exact value of these calibration figures is somewhat irrelevant. The important point is that all radars are calibrated using GPM (single source of reference).

Line 168/169: The older estimate also includes RCA checks or just for radar 63?

As explained earlier in the paper (former ines 115-116), we only have RCA results when the unprocessed radar data are sent to Head Office. That's not the case for this particular radar, so, no this older calibration estimate does not include RCA results. In order to avoid confusion, we have added "… and will assume that its calibration has not changed" for the description of the radar 29 calibration.

Line 173, „Looking at the time series of GPM calibration estimates for other radars than 63 …": Why not refering now to radars 63 AND 29?
This whole paragraph was not well written, thanks for picking that up, and the purpose was not clear. We have rephrased as follows :
" As discussed previously, the RCA technique can be used to accurately track changes in calibration. Unfortunately, among all radars included in Fig. 2, the RCA can only be applied to radar 63. Additional checks of the outputs of the RCA technique for radar 63 (not shown) indicated that the calibration of radar 63 had not changed over that period, which means that we can simply average all the estimates of calibration error from individual overpasses to come up with a more accurate estimate for this radar 63 . Although the RCA technique cannot be used for the other radars, some insights into the the calibration stability can be gained from individual calibration estimates from individual GPM overpasses in each panel of Fig. 2. Considering the expected typical error of 2 dB for individual GPM overpasses as a guideline, it seems reasonable to assume that the calibration of the OceanPOL, Warruwi (77), Dampier (15), Broome (17), and Serpentine (70) radars has not changed over the observational period either, with fluctuations around the mean calibration error estimate less than ~1.5 dB. The Port Hedland (16) radar is more problematic, as the time series shows calibration error estimates ranging from -8 dB to -2.5 dB over that period. However, the three overpass points closest to the date when collocated observations with OceanPOL were collected (26 December 2019) seem to agree reasonably well (around the mean value of -5 dB), so we will use this value of -5 dB in the following but will keep in mind the lower confidence in this calibration figure."

Lines 173ff: So, at the end only radar 16 shows variations? Why not directly writing that instead of starting with 29, then adding 63 and finally the others? Would be easier to read.
Not exactly. Radar 29 is the only one for which we don't have a calibration estimate for the study period. Radar 63 is the only one for which we can directly assess the stability of the calibration using the RCA technique. For all others we just look at the time series fo GPM estimates to assess whether the calibration night have changed. And you are right we only have some doubts about radar 16. We think the paragraph is now clearer, so I hope we addressed your comment.

Line 182, ‚In a perfect world'…: I am not a native speaker, but sounds more like colloquial language to me.
Agreed, we removed that sentence.

Figure 3 caption: Here I suggest to write that the comparison with radar 63 is shown, but in the text you can write that the overall strategy for the OceanPOL comparison with any radar is illustrated using radar 63 in Fig. 3. I also suggest to rewrite the explanation of what is shown in panel b. Maybe something like „…a percentage of all OceanPOL reflectivity values in a resolved 0.5dB bin". And I suggest to write „The number of samples N is 141978 (see panel a)."
Thanks. We have rewritten the text as follows (we have also revised the figure caption) :
" Figure 3 shows an example of the 2D frequency histograms of reflectivity that are used to estimate calibration differences between OceanPOL and any of the radars. This particular figure is for the Berrimah radar (63) for one day (21 November 2019) of the YMCA experiment."

Line 200: Instead of „as on the left panel of Fig. 3)" -> Fig. 3a
Done.

Line 202: Better „Comparison of …provides a better….and allows the detection…"
Done.

Line 206: Better „reflectivities less than 35 dBZ mostly contributed…"
Done.

Line 210: Delete ‚and'
Done.

Line 215, „When including all days of observations for radars 63 and 77": Is there a need to emphasize this? For the other radars not all measurements available are used?
This is explained in sections 2.1 and 2.2.

Lines 217: ‚See' Fig. 4 for ….
Done.

Line 218, „The next best operational radar is radar 70 (Perth).":  Please reformulate.
Done.

Line 289: …derived from all hourly not from all daily estimates, right?
Yes ! Thanks for picking that up.

Lines 304/305: „A major advantage of using a single source of reference is that all radars of the network are calibrated in the same way." This is also the case when other methods are used. If you choose to use the consistency method for your network, you also use the same method for your entire network. Not sure what you want to express here.
We have rephrased as follows: " A major advantage of the GPM VMM technique is that all radars of the network are calibrated against a single source of reference ".
We have not introduced the self-consistency techniques so won't comment on this in the paper, however, we want to stress that the regional variability of the self-consistency technique needs to be investigated before claiming that it can be used with the same level of accuracy to our network that spans several types of weather regimes, from tropical to mid-latitudes.

Lines 305-308: I guess this sentence can be better formulated. Bit hard to read.
Done.

Lines 350ff: For several publications the DOIs are provided, for others not.
This has been homogenised.

---

## Author Response (AR2)

**Responses to reviewers**

**Thanks for taking the time to provide a second review of our paper. Below we respond in red to these latest comments.**

I think the Australian weather radar network is a special case of an operational weather radar network due to the challenge of accessing the network. You should add a sentence on this. Other networks, such as the US or the DWD one do have more routine calibrations (on a six-month basis) and they do not have as large an offset (~ 6 dB) as those observed in your study. In a nutshell, it will be good to state that the observed offsets are not the norm.

Agreed. We have added the following text to address this good point:
"Several radars of the network are installed in very remote locations, bringing specific challenges for the regular maintenance and return to service in case of hardware failure. As a result, maintaining an accurate calibration of this network is more difficult than in other countries. … The internal calibration accuracy of these operational radars is ideally checked six-monthly by BoM radar engineers as part of their routine maintenance. However, periods between visits can be longer for radars in remote locations."

You should make the connection that the BOM network and the ARM network (Kollias, 2019) have challenges in being routinely calibrated, and using spaceborne sensors can help to calibrate them.

Good point. The following comment has been added in the introduction :
" This was also well demonstrated in Kollias et al. (2019) in the context of calibrating the U.S. Atmospheric Radiation Measurement (ARM) cloud radar network using the spaceborne CloudSat radar."

Near the end of the introduction where you added the text: "GPM observations are first used …GPM / ground radar comparisons". Here is a comment: Calibration check requires a multi-source approach in my view: At least as a sanity check, you should use the solar flux measurement @ S-Band to check part of the calibration. Your colleague Mark Curtis has the methods at hand. This should be applied to OceanPol. I think this would add to the paper. But not trying to understand those differences between GPM / surface radars would be a weak spot of this paper in my view.

Well, as the reviewer can see, we share his/her view about requiring multi-source information ! As we explain later in the manuscript, we take every opportunity to do it. However, for most of these radars bar Berrimah and Geraldton, on-site processing removes the sun interferences and ground clutter, so we can only use GPM comparisons for those radars. We had already discussed and included RCA results for Berrimah. Thanks for the suggestion to look at the solar calibration results for OceanPOL, that is indeed a good idea that we had forgotten to explore. This has now been done. Our results show that reflectivities from the sun hits converted to sun power over the study period stays constant at - 93.3 dBm, to within 1 dB. This is a nice confirmation that the OceanPOL receiver calibration has been constant over the study period. The following comment has been added:
" Results using the solar calibration technique for OceanPOL also indicate that the OceanPOL receiver calibration has remained constant, to within 1 dB, over the study period (sun power of about -93 dBm)."

In section 2.2 where you added the following text: "The same calibration procedure as that employed by BoM is used for OceanPOL (internal measurements of gains and losses, no end-to-end calibration)", the reviewer has a few references that will be good to include in the revised manuscript. The references describe how met services do this (you had a comment in the response that you could not find anything). Please add the following references:

V. Chandrasekar and L. Baldini and N. Bharadwaj and P. L. Smith, 2015: Calibration procedures for global precipitation-measurement ground-validation radars. URSI Radio Science Bulletin. doi:10.23919/URSIRSB.2015.7909473.

Frech, Michael, Hagen, Martin, and Mammen, Theo, Monitoring the Absolute Calibration of a Polarimetric Weather Radar, Journal of Atmospheric and Oceanic Technology, 34, 3, 599-615, 2017, DOI: 10.1175/JTECH-D-16-0076.1

Frech, M. and Hubbert, J. Monitoring the differential reflectivity and receiver calibration of the German polarimetric weather radar network Atmos. Meas. Tech.

https://amt.copernicus.org/articles/13/1051/2020/amt-13-1051-2020.pdf

Thanks for these suggestions. The Chandrasekar paper was definitely an omission, and we knew about this paper, it is definitely a reference and an inspiration of our work. We have added it and some text to refer to it : " The extensive recommendations outlined in Chandrasekar et al. (2015) have not been implemented for the Australian radar network yet. "

We have also added the Frech et al. (2017) reference (we did not know about that one, but it's a technique we have used ourselves as well occasionally when disdrometers were deployed atround one of our radars) and the following text :

" The BoM does not operate a disdrometer network. As a result, the technique outlined in Frech et al. (2017), which compares disdrometer simulations of reflectivity with measured radar reflectivities cannot be added to the S3CAR framework. In the future, with the increasing number of dual-polarization radars in the Australian network, we are planning to investigate the benefits of the so-called self-consistency of polarimetric variables and may add this technique to the framework. "

The last reference is about ZDR calibration, which we are not addressing in our study, so we believe it is not relevant here. As a result, we have not added these two publications.